# Human Consumption of Insects in Sub-Saharan Africa: Lepidoptera and Potential Species for Breeding

**DOI:** 10.3390/insects13100886

**Published:** 2022-09-29

**Authors:** Gloria Marceline Numbi Muya, Bienvenu Kambashi Mutiaka, Jérôme Bindelle, Frédéric Francis, Rudy Caparros Megido

**Affiliations:** 1Functional and Evolutionary Entomology, Gembloux Agro-Bio Tech, University of Liege, Passage des Déportés 2, 5030 Gembloux, Belgium; 2Department of Zootechnics, University of Kinshasa, Kimwenza Road No. 01, Commune of Lemba, Kinshasa 012, Democratic Republic of the Congo

**Keywords:** entomophagy, insect food, edible caterpillars, insect industry, food sustainability, host plant

## Abstract

**Simple Summary:**

In the developing countries of sub-Saharan Africa, there is currently growing interest in the consumption of Lepidoptera larvae by humans, due to the important role they play in food security and poverty alleviation. In order to consider Lepidoptera larvae as sustainable alternative protein food, it is important to study the possibility of breeding the species of food interest. A previous literature review on Lepidoptera larvae consumed in Africa revealed a paucity of information on the bioecology of most species and examples of sustainable edible caterpillar rearing practices. This is not surprising given that most research focuses on aspects related to their consumption and nutritional composition. The aim of this work is to collect data on some aspects on their biology, their food plants and provide a guide to orientate the choice of species to raise. In addition, studies on the bioecology and husbandry of edible caterpillars should be more carried out for a sustainable and rational exploitation.

**Abstract:**

There are 472 edible insect species in sub-Saharan Africa, of which 31% are Lepidoptera. Wild harvesting is still the main source of supply for these prized species to this day, with some harvesting techniques negatively impacting the environment. The successful production of edible caterpillars requires the appropriate and efficient implementation of husbandry techniques and practices. In this review, we present current literature on edible caterpillars. We provide a general overview of their life history, nutritional composition, and availability associated with specific host plants, with emphasis on semi-domestication and rearing practices that should replace wild harvest. Based on the assimilated information, a proposal of potential species for farming is provided, with details on key characteristics of development cycles to promote the establishment and development of sustainable farms of edible caterpillars at small and large scales. Such advances would contribute toward reducing anthropological pressure related to the exploitation of these food resources, as well as the environmental footprint of this widespread practice.

## 1. Introduction

By 2050, the world’s population is predicted to exceed 9 billion, which would further aggravate the problems of food security in developing countries. To feed this growing population, food production must increase by nearly 70% and, if possible, double in developing countries. This is because demographic growth in these countries will be coupled with increasing urbanization, and a rise in the middle classes [1]. In a context of increasing scarcity of natural resources and agricultural land, the use of alternative and ecologically sustainable protein sources, including insects (e.g., *Tenebrio molitor*), algae (e.g., *Arthrospira platensis*), and edible mushrooms (e.g., *Psathyrella tuberculate*), seems to be vital to facilitate an increase in world food production [2,3,4,5]. The Food and Agriculture Organization of the United Nations has identified edible insects as one key solution to food insecurity [6].

From a nutritional perspective, insects are not inferior to other protein sources, such as fish, chicken, and beef [7]. In parallel, insect production is considered more sustainable compared to domestic animals, because insects have a high food conversion efficiency, excellent potential (for some species) to be raised using organic by-products [8,9,10], high fecundity, and short development cycles [11]. Edible insects also require less space and water in the process of mass production [12,13,14,15]. For example, one gram of edible protein from beef requires eight to 14 times more land and about five times more water than mealworms [16]. Other more advanced arguments in favor of insects include the fact that they are a rich source of antioxidants and are beneficial for the intestinal microbiota of humans [17,18,19].

In addition to being a nutritious food for some families in developing countries, the exploitation of these non-timber forest products (NTFPs) offers employment opportunities and additional income to people that actively collect, produce, process, and market insects [20]. In developing countries, these edible insects are generally collected in the wild using a wide variety of collection methods depending on the behavior of the targeted insects, as well as the cultures and countries. Methods range from simple hand-picking to the use of specific tools (e.g., glue, sticks, nets, and baskets) [7,21] 

For example, edible caterpillars (e.g., Emperor moth *Cirina forda* (See Table 1 for details of all taxa cited in the text) and African moth *Imbrasia oyemensis*) are usually harvested by hand [22,23]. Termites (*Macrotermes natalensis*) and grasshoppers (*Ruspolia differens*) are attracted using light traps. Dragonflies (*Orthetrum sabina*) are collected using sticky sap from fruit trees or glue spread on the end of long poles [24]. Orthoptera, such as *Brachytrupes membranaceus*, are easily detected based on the sound they emit, and are captured by hand [7,24].

Although recognized as the primary source of edible insects, natural environments only offer seasonal production, with limited availability. The increasing and continuous demand for edible insects has led to an imbalance in the forest ecosystem, with repercussions on the survival of certain insect species and/or their host plants and natural predators [24,25]. 

Thus, the development of breeding methods would facilitate a continuous supply of edible insects, with a reduced impact on the environment [7]. In parallel, the controlled production of insects would help reduce health risks associated with their consumption, by avoiding the possible bioaccumulation of substances harmful to the health of consumers from certain toxic plants, polluted areas, or farmed areas containing pesticides [26]. In Africa, very few species are mass produced. Examples of mass produced insects include silkworm caterpillars of *Bombyx mori* and *Gonometa postica palmarum* [24]. This almost non-existent production of edible caterpillars should raise interest in developing both caterpillar breeding systems and silvicultural systems aimed at multiplying the host plants that serve as food to insects in the vicinity of homes, villages, and farms. Such practices would involve local communities in the management of these initiatives and protection of targeted species [22,27].

Worldwide, more than 2 billion people consume more than 2000 species of edible insects, occasionally to regularly. Targeted insects mainly belong to eight of the 12 insect orders [11,28,29,30]. In descending order, the most consumed insects worldwide are Coleoptera (beetles), Lepidoptera (caterpillars), Hymenoptera (bees, wasps and ants), Orthoptera (grasshoppers, crickets), Hemiptera (cicadas, leafhoppers, mealybugs and bugs), Isoptera (termites), Odonata (dragonflies), Diptera (flies) and some species of insects belonging to other orders [6]. Among these orders, Lepidoptera have the greatest diversity of species consumed in tropical Africa. Lepidoptera represent 31% of edible species out of 472 identified insect species (Figure 1), belonging to 128 families, of which 36 include species consumed by humans, mainly in the form of caterpillars and, more rarely, chrysalides [31].

Thus, this literature review focuses on demonstrating that caterpillar consumption is a common practice in tropical Africa, and that this traditional food source deserves greater attention. We review information on the biology of edible Lepidoptera, their food value, appreciation by local populations, their food plants, and the environmental risks associated with their exploitation. We apply this information to identify potential avenues for species domestication programs, and potential issues associated with edible caterpillar farming.

## 2. Life Cycle of Lepidoptera

Published literature documenting the developmental cycle of tropical Lepidoptera remains limited (Table 2). Most species widely consumed in Africa produce a single generation per year (e.g., *C. forda*). This phenomenon is thought to be regulated by abiotic factors, such as photoperiod, temperature and host plant availability, which mainly affect pupation [32]. However, species such as the caterpillars *Imbrasia belina*, *Bunea alcinoe* and the African moths *Gonimbrasia zambesina*, *Gonimbrasia krucki*, *Gonimbrasia cocaulti*, and *Gynanisa nigra* can complete two cycles in one-year while others, such as the Eri silkworm complete several cycles in a single year [30,33,34,35].

## 3. Nutritional Composition of Edible Caterpillars

Insects are an alternative food source that has a high content of essential nutrients (proteins, lipids, and minerals) for humans and animals [43].

Malaisse [44] provides a detailed overview of the nutritional values of some edible caterpillars from sub-Saharan Africa, confirming the empirical knowledge of local populations. The nutritional analysis of 24 species of dried edible caterpillars allowed us to determine the average proportion of proteins (63.5%), lipids (15.7%), and energy value (457 kcal/100 g) contained in these insects on a dry matter basis [22,45]. These data revealed clear variations in nutritional composition of different edible caterpillars. This variation is associated with species, stage of development, biotope, diet, method of preparation (e.g., roasted or boiled caterpillars), and analytical method used (Table 3, [46,47]). Edible caterpillars have higher protein levels (28 g/100 g on a fresh matter basis) compared to chicken meat (21 g/100 g of fresh matter protein). The energy intake of caterpillars (370 kcal/100 g) is similar to pork (416 kcal/100 g) [10].

Although a 100 g portion of insects is not enough to ensure the daily vitamin needs for humans (e.g., A and C), they also contribute different vitamins (depending on species) (e.g., thiamine/B1, riboflavine/B2, pyridoxine/B6, pantothenic acid, niacin) and minerals (e.g., K, Ca, Mg, Zn, P, Fe). Their bioavailability provides a means of combating malnutrition in Africa and preventing metabolic diseases [4,22,48,49,50,51,52]. Because of their high nutritional value, caterpillars are sometimes mixed with flour to prepare a porridge at breakfast to combat malnutrition in children, frail people, and pregnant women [4,45,53,54].

Proteins are major nutritional components of insects, providing essential and non-essential amino acids to the human body [43,55]. The digestibility of insect proteins is comparable to that of casein or soy proteins (77–98%) [43]. Some studies have reported that the digestibility of the moth *Clanis bilineata* was 95.8% compared to casein [51]. Oibiokpa et al. [48] also showed that the moth *C. forda* has a higher biological value (86.90%) compared to casein (73.45%).
insects-13-00886-t003_Table 3Table 3Nutritional composition of edible caterpillars consumed in Africa (dry matter basis).FamilySpeciesProtein(g/100 g)Carbohydrates(g/100 g)Fat(g/100 g)Ash%Energy(kcal/100 g)Fiber%LocationNotodontidae*Anaphe infracta*20/15.21.6/2.4Guinea-Conakry, Cameroun, DRC, Equatorial Africa*Elaphrodes lactea*53.61.421.93.9–4.2417/Zambia, DRCSaturnidae*Bunaea alcinoe*74.3/14.12.9//Mali, Burkina Faso, Nigeria, Ghana, Cameroon, South Africa, Zambia, DRC, Congo, RCA, Zimbabwe, Nigeria, Tanzania*Cirina butyrospermi*631314.55.14325Togo, Mali, Burkina Faso, Nigeria, Ghana*Cirina forda*12–74.41.7–75.3–20.21.5–11.5359–4105.2–9.4Nigeria, DRC, CAR, Zambia, South Africa, Botswana, Burkina Faso, Mozambique, Namibia, Ghana, Togo, Tchad, Cameroon*Imbrasia epimethea*57.8–730.9–5.312.4–233.3–7.5419–501/Guinea-Conakry, DRC, Zambia, South Africa, Cameroon, Congo, CAR, Zimbabwe*Imbrasi belina*35.2–56.97.8–10.910–23.36.9–11.340227.8DCR, Zambia, South Africa, Zimbabwe, Botswana, Malawi*Imbrasia oyemensis*23.7–61.61.5–1119.1–57.72.6–5.5384–477
Guinea-Conakry, Cameroon, DRC*Imbrasia obscura*62.3/12.2///DRC, South Africa, Zimbabwe, Botswana, Gabon, Mozambique, Namibia*Imbrasia truncata*54.5–731.2–1115.2–27.82.7–5.5418–499/DRCSource: [40,43,49,50,53,56,57,58,59,60,61,62].


However, the digestibility of insect proteins could be improved by eliminating the rigid chitin-rich exoskeleton, which reduces the digestibility of their crude proteins, despite the presence of two chitinases in the human stomach [48,63,64]. Chemical methods can be used to remove the exoskeleton, whereby strong acids and bases are used to dissolve calcium carbonates and proteins, respectively [65]. For example, chitin removal by alkaline extraction increases the digestibility of bee protein from 71.5% to 94.3% [63].

The quality of proteins depends on the amino acid composition [66], with insects being particularly rich in lysine and threonine, but sometimes deficient in methionine and cysteine [67]. However, some amino acids could be limiting depending on insect species (Table 4, [49,68]). Many amino acid sequences have been identified in a wide range of dietary proteins that are generally considered to be sources of bioactive peptides (BAPs), such as the tripeptides valine-proline-proline (VPP) and isoleucine-proline-proline (IPP), and the polypeptides phenylalanine-phenylalanine-valine-alanine-proline- phenylalanine-proline-glutamate-valine-phenylalanineglycine-lysine (FFVAPFPEVFGK) and tyrosine-leucine-glycine-tyrosine-leucine-glutamate-glutamine-leucinearginine (YLGYLEQLLR) [69,70]. These BAPs might have biological functions and hypotensive, antioxidant, antidiabetic, immunomodulatory or mineral-binding properties [69,70]. Among known edible insect species, the biofunctional properties of proteins and peptides from *B. mori* have been extensively studied. For example, analysis of angiotensin converting enzyme (ACE) revealed the existence of angiotensin I converting enzyme (ACE) inhibitory peptides that reduce blood pressure. For example, peptides identified in the protein of *B. mori* pupae include tripeptides (KHV and ASL) and the pentapeptide GNPWM [71,72,73].

Reference protein intake for health adults is estimated to be 0.83 g protein/kg body weight per day (i.e., 62.3 g for a 75 kg adult). Fogang et al. [74] reported that the consumption of a 100 g portion of *Imbrasia truncata* or *Imbrasia epimethea* caterpillars covered 30.6% and 32.3% of the required protein intake of a 75 kg adult, respectively.

Lipids are the most energy-dense group of macronutrients [2]. They store and provide energy, and support and protect the various organs [83]. They are made up of triglycerides, each with a glycerol molecule and three fatty acids that are saturated or unsaturated [2]. Caterpillars are among the most fat-rich insects [84].

Like other edible insects, edible caterpillars are a source of fatty acids [45], with most caterpillars being rich in mono- and, even, polyunsaturated fatty acids (PUFA) [85]. These fatty acids are mainly linolenic (C18:3n3) and linoleic (C18:2n6) acids, commonly known as omega-3 and omega-6 fatty acids, respectively [85]. These PUFAs are not synthesized by the human body, and must be obtained through the diet [65]. Therefore, the ingested amounts and balanced proportions of these fatty acids should be provided sufficiently, as unbalanced ratios are often associated with health problems in humans, such as coronary heart disease, cancer, and autoimmune and inflammatory diseases [86].

Table 5 shows the fatty acid composition of some caterpillar species consumed in sub-Saharan Africa. The fatty acid composition of the lipid content of *C. butyrospermi*, *C. forda* and *I. belina* caterpillars is estimated to contain 37–54% of PUFAs, 32–57% of saturated fatty acids (SFA), and 1–27% of monounsaturated fatty acids (MUFA) [50,52,87]. Other edible lepidopteran larvae (such as confused Emperor *Imbrasi ertili* and *Heliothis zea*) are composed of up to about 70% unsaturated fatty acids [88]. There is large variability in the fatty acid profiles of different caterpillar species. These differences in fatty acid composition are attributed to different fat and calorigenic content, along with the diet of caterpillars [45]. 

The diets of people living in developing countries are generally characterized by micronutrient deficiencies, resulting in major health consequences [7]. Interestingly, the micronutrient content of edible insects is influenced by their diet [91]. Importantly, the consumption of edible insects would provide significant amounts of minerals that are sufficient to meet human needs. Examples include copper (e.g., *Usta terpsichore*, mealworm adult), iron, and zinc (e.g., *B. mori*), as well as vitamins (carotene and vitamins B1, B2, B6, D, E, K, and C) [51].

Certain minerals (such as iron and zinc) are of particular interest, because they are often the source of deficiencies in developing countries [7,11]. Fe and Zn deficiency is particularly prevalent in regions with high cereal and low animal food consumption. In fact, both Fe and Zn help prevent malnutrition and early stunting [92]. Mwangi et al. [92] compared the Fe and Zn content in meat from conventionally raised animals (8 mg/100 g Fe and 21 mg/100 g Zn beef, 4 mg/100 g Fe and 6 mg/100 g Zn pork, and 3 mg/100 g Fe and 6 mg/100 g chicken) against three edible insect species (6 mg/100 g Fe and 13 mg/100 g Zn *T. molitor*, 14 mg/100 g Fe and 21 mg/100 g Zn *Acheta domesticus*, and 19 mg/100 g Fe and 15 mg/100 g Zn *L. migratoria*). The authors showed that edible insects contained Fe and Zn levels similar to, or higher than, those of conventional farm animals [92]. Of note, mineral content varies according to insect species, stage of development, and diet [7,43]. Better control of food intake would enable easy modification of insect mineral content [11].

The presence of iron in edible caterpillars (*Cinabra hyperbius*) allows could eliminate iron deficiency anemia in people that consume them, due to the lack of protein necessary to synthesize blood cells, which is an issue some children have in developing countries [45]. The iron content of *C. butyrospermi*, *C. forda*, *I. epimethea*, *I. belina* and the African moth *I. truncata* caterpillars ranges from 1.30 to 64 mg/100 g dry weight. These values are higher compared to beef, which is 6 mg per 100 g dry weight. Similarly, Zn levels in *C. forda* and *I. belina* caterpillars range from 3.71 to 24.2 mg/100 g dry weight, whereas beef contains an average of 12.5 mg per 100 g dry weight (Table 6, [93]). 

Although 100 g of edible caterpillars does not meet the daily calcium requirement, with the exception of the edible caterpillar (*Tagoropsis flavinata*) [45]. Mineral requirements could be met by consuming edible caterpillars, meeting recommended nutrient intake for adults (Table 6). Rumpold & Schlüter [43] reported that some caterpillars (*U. terpsichore, Imbrasia ertli)* have high levels of sodium per 100 g and, in some cases, could exceed the maximum daily intake of 1500 mg. In parallel, insects are sources of vitamins [22]. As with mineral content, the vitamin content of insects could be altered if they are raised on vitamin-rich substrates [91]. Of note, data available on vitamins present in edible insects varies across publications [2]. These differences are likely attributed to the species used, analytical methods, and insect diet [51]. Depending on the species, caterpillars are rich in various water-soluble vitamins (thiamine/B1, riboflavin/B2, pyridoxine/B6, pantothenic acid, niacin). According to Malaisse [44], the daily consumption of 50 g dried caterpillars would meet human needs for riboflavin and pantothenic acid. Furthermore, 30% of niacin requirements would be met [22].

## 4. Availability, Host Plants, Harvesting, and Storage

### Availability and Relationship of Lepidoptera with Host Plant(s)

The seasonal availability of edible caterpillars varies with region and reflects variation in climatic conditions [16,22,30]. In the Central African Republic (CAR), caterpillars are available from mid-June to late September [20], whereas they are available from July to October in Cameroon and from August to January in Congo-Brazzaville. In the Democratic Republic of Congo (DRC), edible caterpillars are available between July and September in the western Kasai region, between June and September in the Kisangani region [20], and from September to December in the Bandundu region [20]. 

This seasonality is related to the presence of plants on which caterpillar feed at the beginning of the rainy season, because caterpillars specifically feed on one or more host plants that only grow in certain ecosystems [30,94]. Several studies have provided information on the host plants of edible caterpillars in tropical Africa [27,94,95,96,97,98]. These studies show that edible caterpillars are generally polyphagous, associating with several host plants (Table 7). For example, the caterpillar *C. forda* feeds on *Vitellaria paradoxa* (Sapotaceae) in West Africa, *Autranella congoensis* in CAR, and *Burkea africana* in South Africa. In the DRC, these caterpillars are associated with *Crossopteryx febrifuga* in Bas-Congo, *E. suaveolens* in the Kisangani-Tshopo region, *Erythrophleum africanum* in Bandundu, and *Albizia antunesiana* in Katanga [99].

Of note, the abundance and availability of edible caterpillars is sometimes affected by the felling of host plants. For example, the woody plants, Sapelli (*Entandrophragma cylindricum*) and Tali (*Erythrophleum suaveolens*) are both widely harvested for timber and edible caterpillars (*I. oyemensis* and *C. forda*, respectively). Consequently, conflict has risen between one-time timber harvests and the annual harvests of edible caterpillars spanning decades [105]. Forest management approaches should support the production of these wood and non-wood resources to benefit multiple stakeholders in these forests. This would minimize potential conflicts between logging and the needs of local people who consume these wild foods [105].

To improve the management of these resources, it is important to provide information on the yields of edible caterpillars inhabiting timber trees, and how logging affects their availability. For instance Muvatsi et al. [105] quantified the density of Sapelli and Tali trees of different size classes within a 10 km radius of four villages in the Kisangani region (DRC) in 2012, along with the annual cutting areas of two logging concessions. Stumps of these two forest species were identified and measured in 21 five-hectare plots around each village and 20 five-hectare plots in each concession. Around the villages and in the concessions, Sapelli were present at densities of 0.048 ± 0.008 harvestable trees (≥80 cm diameter at breast height [dbh]) ha^−1^ and 0.135 ± 0.019 precommercial trees ha^−1^. Harvestable Tali trees (≥60 cm dbh) were seven times more abundant with 0.347 ± 0.032 ha^−1^, while precommercial Tali trees were present at densities of 0.329 ± 0.033 trees ha^−1^. Based on estimated tree densities, caterpillar yields were estimated for a 15,700-ha semicircle within a 10 km radius of the villages. Depending on the village, yields were estimated at 11.6–34.5 kg yr^−1^ for *I. oyemensis* on Sapelli trees, and 65.8–80.9 kg yr^−1^ for *C. forda* on Tali trees, averaging 0.74–2.2 kg ha^−1^ yr^−1^ and 4.2–5.2 kg ha^−1^ yr^−1^ fresh weight, respectively (0.23–0.68 kg ha^−1^ yr^−1^ and 1.3–1.6 kg ha^−1^ yr^−1^ dry weight, respectively).

## 5. Harvest

Edible caterpillars are harvested in forests, savannahs, and other uncultivated lands based on the periods and harvesting sites known to farmers [106,107]. Prized species are identified from accumulation of their droppings at the base of tree trunks where they feed on leaves, their characteristic smell, or consumed leaves [20]. In general, caterpillars are harvested manually by women and children on the ground, trunks, branches, or leaves of plants on which they are found [108,109].To reach caterpillars at the top of large trees, some harvesters hit the trunks with hammers, or use a long bamboo pole to knock the caterpillars off the top of the trees, while some pygmy ethnic groups in the CAR climb trees to harvest them [20]. Caterpillars are also harvested by felling host plants. For example, in the DRC, 65% of respondents stated that they cut down trees to harvest the most commonly consumed caterpillar species (including *C. forda*, *I. epimethea*, *I. ertli*, and *I. oyemensis*) despite the threat to the ecosystem and targeted caterpillar species [25].

The sanitary quality of caterpillars collected from the natural environment raises some public health concerns because it is difficult to control the environment in which they are found and the level of toxins transferred from the host plants on which they feed, along with the possible risks of microbiological contamination [7,40,110].

The bibliographical syntheses of Rothschild et al. [111] and Nishida [112] exhaustively list the Lepidoptera that potentially accumulate various toxic substances, including cardenolides, aristolochic acids, HCN, histamine, acetylcholine, cardiac glycosides, and pyrrolizidine. Some edible caterpillars contain antinutritional factors, including phytate, tannin, oxalate, hydrocyanide, saponins, and alkaloids. These antinutritional factors affect and inhibit the availability of food nutrients [113,114]. For example, antinutritional analysis of the degutted, boiled, dried, and ground larvae of *C. forda* collected in the wild revealed the presence of oxalate (4.1 mg/100 g) and phytic acid (1.0 mg/100 g), whereas tannins were not detected [113]. Similarly, antinutritional analysis of *B. alcinoe* larvae revealed the presence of oxalate (15.47 ± 1.88 mg/100 g), phytate (18.21 ± 2.14 mg/100 g), and cyanide (1.68 ± 0.20 mg/100 g), which were within the permissible limits [115].

Some edible caterpillars (African silkworm *Anaphe venata*, *B. mori*) also exhibit thiaminase I activity. This enzyme induces thiamine deficiency or beriberi [116,117]. The thiaminase activity of *B. mori* caterpillars is less than one-third that of *A. venata* [51]. In parts of southwestern Nigeria, an acute, seasonal ataxic syndrome characterized by tremors, ataxia, and variable levels of altered consciousness has been observed annually for over 40 years after consuming slow-cooked, high-thiaminase chrysalises or silkworms of African silkworm, *A. venata* [51,117]. The characteristics of *Anaphe* thiaminase include high heat resistance, which explains why symptoms of seasonal ataxia appear within hours of eating a carbohydrate meal with a stew containing roasted *A. venata* larvae [116]. Detoxification of the African silkworm would make these prized larvae a safe source of food for local populations [51,117]. Some studies have also reported the presence of bacterial populations associated with the exoskeleton and intestinal contents of *B. alcinoe* larvae freshly collected from the wild in Nigeria [40,113]. Specifically, nine bacterial isolates belonging to the genera *Staphylococcus*, *Bacillus*, *Micrococcus*, and *Acinetobacter* were isolated from the exoskeleton of *B. alcinoe*. Furthermore, 11 isolates representing the genera *Staphylococcus*, *Bacillus*, and *Micrococcus* were recovered from the intestinal contents of *B. alcinoe*. Importantly, this study demonstrated that *Staphylococcus aureus* was present both on the exoskeleton and in the intestinal contents, along with a *Bacillus* species (*Bacillus cereus*), both of which produce enterotoxins. However, while *S. aureus* can be destroyed at high cooking temperatures, *B. cereus* has the ability to resist such temperatures via the production of endospores. Therefore, post-harvest treatments are essential to prevent any contamination or deterioration of products. Treatments could include screening of marketable insects, development of appropriate treatment standards, and the manipulation and storage of collected insects. Such actions would minimize all associated risks with consuming insects, ensuring the safety of consumers [113,118].

In addition to their role in feeding households, harvested edible caterpillars also contribute toward improving the economy of local communities by marketing part of their harvest [22]. The edible caterpillar trade generates additional income within the households that practice it, and contribute significantly toward improving the living conditions of local populations [109]. Caterpillars can be marketed both directly and indirectly; however, in most cases, sales are completed indirectly. The direct method involves no intermediaries between the producer (collector) and consumer, whereas indirect sales involve intermediaries (wholesalers and resellers) between the collector and consumer [20]. However, the method of selling caterpillars individually is only profitable for producers. For example, in Cameroon, the local population makes significant profits, and earns much more than the salary of a servant, gardener, or even a driver. The sale of edible caterpillars provides wholesalers and retailers with a profit margin of 125% and 21%, respectively [20]. In the DRC, the caterpillar trade provides employment opportunities for the Congolese, providing opportunities to earn profit margins of over 100% [97]. Vantomme et al. [22] reported the import of about 8 t of dried *Imbrasia* sp. to France and Belgium from the DRC, valued at about US$41,500 in Belgium (average price of US$13.8 per kg). Improving how the caterpillar market is organized would facilitate a move from individual sales to contractual sales, which would benefit all actors in the production chain (i.e., producers, intermediaries, and consumers) [97]. 

In the past, the harvesting of edible caterpillars was regulated by: (a) monitoring host plant abundance and possible changes to the ecosystem; (b) protecting vulnerable life stages and specific habitats (e.g., timing of forest fires to reduce the destruction of host plants and/or eggs and caterpillars); and (c) determining a harvesting schedule. Unfortunately, population pressure, poverty, and increased demand for caterpillars from outside buyers have led to changes in human behavior [16]. Certain edible caterpillar species have been overexploited, including the mopane caterpillar (*I. belina*) in the Miombo forests of southern Africa [25] and Sapelli caterpillars (*I. oyemensis*) in Central Africa. The excessive collection of these caterpillars has adverse effects on future harvests [22]. The availability of edible caterpillars might also be affected by logging, as the host trees of some edible caterpillar are also used as valuable timber (e.g., *E. cylindricum*) or artisanal charcoal production [105,119,120]. Deforestation destroys the natural habitat of edible caterpillars and their host plants by reducing the amount of foliage used to feed them [121]. Conversely, overexploitation, forest fires, and overgrazing have caused the extinction of many caterpillar species, threatening the well-being of local human populations, especially highly vulnerable rural populations [20,122,123]. Finally, edible insects are also part of the food chain of other animals, including birds and other small vertebrates. Therefore, any decrease in edible caterpillars would also affect the stocks of their predators, threatening species biodiversity [22].

Conversely, the harvesting of edible caterpillars positively feeds back on bushfire frequency and forest management [22,124,125]. Fires set at the end of the dry season, when the weather is too hot and trees start to produce new leaves, causes widespread damage by killing trees, reducing regrowth, and increasing erosion [124]. Some studies show that late fires are almost absent in areas where edible caterpillars are harvested by villagers, inducing favorable impacts on forest management [22]. Harvesting caterpillars encourages people to burn early, which protects the caterpillars and promotes forest regeneration [124]. Therefore, it is more beneficial to have early fires, which helps avoid damage caused by late fires [125]. Likewise, Leleup and Daems [126] identified the relationship between brush burning dates and the seasonal occurrence of different developmental stages of certain edible caterpillar species in the DRC. Based on this information, these authors provided several recommendations on optimal periods for burning savanna and open forest [22]. 

Since most edible caterpillars are harvested from the wild, the development and adoption of harvesting methods appears to be the simplest conservation management option [107]. Evaluation of each species should include the determination of biological factors, including distribution, host plant (or habitat) specificity, range of host plant(s), various life history parameters (number of generations, developmental time), dispersal ability, and abundance. The suitability for the semi-domestication or breeding of these species should also be investigated. Emphasis should be placed on establishing sustainable harvesting protocols, including the possibility of using simple semi-domestication procedures to breed local species [20,107]. In addition, to contributing to the sustainable production of edible caterpillars, silvicultural systems that integrate forest management by supporting the multiplication of tree species (e.g., *Entandrophragma* spp., *Isoberlinia* spp., and *Colophospermum* spp.) associated with caterpillars of high food value could be developed [127]. Thus, local communities could benefit from both caterpillars and wood production, while providing an opportunity for harvesters to better protect and manage the habitats from which these insects are harvested [22,27]. For example, in Central Kongo Province (DRC), *Acacia auriculiformis* is planted for firewood, but also supports five species of edible native caterpillars [99]. 

## 6. Conservation

After harvesting, some caterpillars can be kept alive for about three days, or dried for a shelf life of several months. In the latter case, the preservation treatment consists of exposing caterpillars to the sun or simply smoking them [20]. Larvae are dried in the sun on wooden grids or corrugated sheets, with a long period of sunlight being required. This technique is less practiced than smoking, because the shelf life is shorter. Before exposing caterpillars to the sun, they are eviscerated shortly after they have been removed from the foliage of host plants, roasted in hot ashes from wood fires to remove their hair until they harden, and, finally, they are exposed to the sun until they become crispy and dry [20,95]. A smoking step extends the preservation of caterpillars to a period of three months [20]. It involves spreading the insects on a grate (wooden or other substrates) placed over a heat source. The caterpillars are frequently smoked after being boiled for 30 to 45 min [20]. Therefore, the food safety of insects harvested in the wild and the method of preservation requires evaluation.

## 7. Semi-Domestication and Farming

Wild harvesting, semi-domestication, and farming are the three possibilities for obtaining edible insects. Of the recognized edible insect species, 92% are harvested from the wild, 6% are considered semi-domesticated, and only 2% are considered farmed [107]. Despite the small percentage of insects that are produced or semi-domesticated, these production methods represent huge potential for the sustainable supply of edible insects, which would reduce the negative impacts of wild harvesting on the natural environment [107,128].

### 7.1. Semi-Domestication

The domestication of edible insects appears to be a relatively simple and economical method of production. However, to date, only silkworms (*B. mori*) and honeybees (*Apis melifera*) are considered fully domesticated [129]. 

For an edible insect species to be targeted for domestication, a number of criteria should be met, including a short reproductive cycle, gregarious behavior, high reproduction and survival rates, modifiable diet, and high nutritional value [124,128]. For this purpose, *S. ricini*, would be a suitable candidate. It is a multivoltine species with at least six to seven generations per year. It has a total life cycle duration of 53 and 57 days for females and males, respectively. It has a hatching rate of 97%, with highly fecund females (360 eggs on average) [130].

Very few examples of semi-domestication of edible caterpillars are available. Semi-domestication involves the transfer of insect specimens to other breeding sites or the manipulation/creation of habitats [27]. Some first instar caterpillars of the edible Saturnidae (e.g., *Lobobunaea phaedusa*, *I. obscura*, *I. eblis*, and *I. epimethea*) are transferred from wild trees in forests to domestic trees near to the dwellings of villagers [21]. This action allows larvae to continue their development in an environment where they are kept safe from drought, heat, and predation, all of which contribute to their mortality in the wild [6,21]. In Bas-Congo, the caterpillars *I. obscura*, *I. ertli*, and *C. forda* are introduced to villages by placing them on trees close to the houses of farmers and on their lands [99]. Malaisse [44] also recorded that branches with many caterpillars are transferred to the same tree species closer to villages. This system allows villagers to monitor the caterpillars and improve the harvesting schedule. If the area is protected from fire during this phase, the moths of some species lay their eggs on the tree beneath which they previously pupated, or on nearby trees. One farmer followed this facilitation system by introducing *C. forda* caterpillars to a savanna area abundant in the host tree species *C. febrifuga*, and subsequently harvested caterpillars regularly following introduction [131]. Other authors, such as Ngoka et al. [132], reported cases of semi-domestication with wild African silkworms *Gonometa postica*. However, it is often difficult to semi-domesticate the species of *Gonometa* spp., with failures being recorded for *Gonometa rufobrunnea* [132,133]. This failure has been attributed to several relevant questions remaining unanswered. These include the factors that induce or cause diapause, conditions required for successful mating, development of effective methods for seeding populations, and the conditions required for the successful mating of butterflies. Therefore, further studies are needed to determine the appropriate captive rearing and breeding conditions for this species to enable sustainable production [133]. For instance, an initial population of *G. postica* live pupae in cocoons were reared under semi-captivity after collection from host plants in the study area. Oviposition occurred on the branches of its two host plants (*Acacia Mearnsii* and *Acacia hockii*), which were protected by netting to protect the larvae from predators and parasitoids. This rearing technique allowed the development cycle of this species to be characterized (i.e., egg incubation of 14 to 18 days, larval development of 73 to 99 days, and pupal development of 48 to 72 days).

Ghazoul [134] reported the potential of rearing a population of *I. belina* in captivity for three years. However, the long-term viability of continuous production of fresh mopane worms needs to be ascertained, along with the control of viral or parasitoid disease accumulation, particularly when worms are unavailable for long periods. High mortality might occur as a result of unprotected natural conditions, exposing worms to disease, parasitoid attack, predation, heat, and drought. However, such mortality could be controlled and mitigated by introducing various approaches, such as the use of protective sleeves around branches and the construction of shade houses. Larvae can be successfully reared in shade cloth bags draped over small trees, especially until the end of the second or third instar [134].

Under normal environmental conditions, larvae can be effectively released from bags at the end of the second instar (or from the greenhouse during the third instar) on exposed trees with fresh leaves. After this point, mortality is relatively low; thus, worm production could be increased, without needing to invest effort in protecting large 4th and 5th instar larvae. In contrast, under persistent hot and dry conditions, larvae placed in shade cloth bags overheat and dry out. In such cases, larvae could be reared successfully in larger shade bags under dry conditions; however, the risk of disease remains, which appears to accumulate over the season. Unfortunately, under both conditions, the risk of increasing viral load remains a major issue. This risk is attributed to the fact that larvae are maintained for long periods in the same area over multiple years. Thus, viral infection appears to be the main factor limiting mopane worm production, along with pupal diapause [134]. 

### 7.2. Farming

Although insect farming for human consumption has been introduced in both tropical and temperate countries, it remains a relatively rarely used method of producing edible insects [128]. In practice, insect farming is not very common [128]. 

A major challenged face by breeders in rearing edible caterpillars is the lifting of pupal diapause and ensuring the sufficient production of food plants for caterpillars. For example, in Zimbabwe, the second generation of mopane worms undergoes a long diapause of six to seven months. This presents an opportunity for scientists to break the diapause to allow the year-round supply of mopane worms. The physiological processes that *I. belina* undergoes during diapause, however, are not yet fully understood or documented, with only general information. For most insects, these processes are primarily regulated by abiotic factors, such as photoperiod and temperature. It is important to understand how photoperiod and temperature affect these processes to predict the initiation and termination of diapause in the field [32]. Two successful pupation methods include boxes and pits. In the first method, plywood boxes with hinged lids were constructed with dimensions of 26 × 52 × 32 cm. To evaluate both the suitability of substrate and survival of larvae in boxes, different types of soil were used, filling up to half or three-quarters of the height of boxes. Then, different numbers of caterpillars at the prepupal stage were placed in the boxes. Some mopane leaves were included in case some larvae were still feeding. After about one month, the boxes were emptied to count the number of live, deformed, dead, and parasitized pupae. In the second method, open pits were constructed by digging a 2.5 × 1.0 m area to 20 cm depth. Sheets of plywood were placed along the sides of the pit. The pit was then covered with shade cloth. Each pit was filled with soil to a depth of about 20 cm to hold the plywood sheets in place, which extended about 25 cm above ground level. Then, the whole structure was covered with a shade cloth. Finally, pre-pupating larvae were placed on the surface of the pit with some fresh mopane leaves [134].

Both methods could be used; however, the pit method is less labor intensive and less amenable to experimentation and monitoring. Pupae remain viable for at least 2.5 years, and artificial cooling prolongs adult hatching. To continue mopane worm rearing, the conditions under which dormancy could be extended or adult emergence artificially induced require investigation [134]. 

There are very few cases of edible caterpillar breeding because breeding requires suitable mastery of the ecology and biology of targeted species, both animal and host plant. However, various laboratory rearing trials have been conducted in Africa, starting with emblematic caterpillars, including *C. forda* and *C. butyrospermi* [33,37]. For example, for *C. butyrospermi*, rearing involved feeding caterpillars individually in Petri dishes with young shea leaves given *ad libitum*, and then the different biological parameters were measured. A developmental time from egg to adult of approximately 398 days was recorded, with egg incubation of 30 days, larval development of 33 days, and a very long pupal stage of 335 days. In contrast, adults lived just 5 days [37]. In the Lwiro region of the DRC, the production of *A. infracta* fed ad libitum with the leaves of their host plant *Bridelia micrantha* resulted in the production of 9.32 kg caterpillars based on 10 pairs of adults over a 49-day feeding period [39]. For this species, the pupation time was relatively short (77 days), which is an advantage for the mass production of caterpillars. In addition, in the DRC, Muya et al. [42] described the development cycle of the caterpillar *Aegocera rectilinea*, locally called “Mikombidila”. Again, the caterpillars were fed *ad libitum* with the leaves of their host plant, *Boerhavia diffusa*, with the development cycle from egg to adult lasting about 36 days. Due to this short cycle, it would be possible to achieve about 10 cycles per year under controlled conditions for this multivoltine species. Thus, the profitability and feasibility of Lepidoptera rearing would benefit by focusing on multivoltine species. To produce edible caterpillars effectively, short-cycle species (e.g., *A. rectilinea*) should be targeted, until the issue of long pupal diapause, which seems to be a characteristic of most edible caterpillars in the family Saturnidae, is solved (Table 2).

## 8. Conclusions

This review consolidated available data on edible caterpillars in sub-Saharan Africa regarding both ecology and rearing. First, information was assimilated on their life cycle, nutritional composition, availability, and relationship with the host plant(s). Then, methods and issues were considered with respect to promoting breeding by optimally controlling their development cycle and the production of their food plants to obtain food and nutritional security. Emphasis was placed on identifying caterpillar species with a high potential for breeding via mass production to ensure a sustainable supply.

To date, most research on edible caterpillars has focused on their diversity, collection, preparation, consumption, nutritional composition, and host plant diversity. However, there is a paucity of studies on the life history and rearing techniques of prized species.

More research is needed to elucidate the bioecology of Lepidopteran life cycles. Experimental rearing could be used to determine various aspects related to their biological parameters. For instance, it is necessary to measure the duration of each stage of the development cycle (egg, larva or caterpillar, nymph or chrysalis, adult or butterfly), type of pupation (underground and/or aerial), the longevity of adults, rate of fecundity, reproduction and hatching, survival rate of larvae and chrysalises, optimal conditions for reproduction, and embryonic and larval development.

Ultimately, to reap the benefits of edible caterpillars at scale, rearing technologies must be intensified rather than depending on wild harvesting. This step, in advance, requires the selection of potential species for farming. Criteria that should be considered include their suitability for mass production, biomass supply, nutritional content, and environmental implications. Thus, it is important to invest in the research and development of breeding techniques for edible caterpillars. These caterpillars have high biological and nutritional value as a potential source of protein, in addition to providing various socio-economic benefits to local communities in Africa. We suggest that future research begins with identifying Lepidoptera species of which the larvae and caterpillars are consumed, but for which there is limited information on their bioecology. Based on Table 2, we propose five caterpillars (*A. infracta*, *A. rectilinea*, *C. forda*, *C. butyrospermi*, *E. lactea*, and *I. belina*) that should be investigated to integrate into production systems, as they are among the most consumed and valued caterpillar species. Among these species, the caterpillars *A. infracta*, *A. rectilinea*, and *I. belina* have more advantages in terms of characteristics related to seasonality and length of the development cycle (Table 2). Table 2 also shows that information on the life cycle of certain edible caterpillar species (*B. alcinoe*, *I. obscura*) is extremely limited. By collecting these data, future studies could help characterize the life cycle of these species and the phenology of associated host plants. Future studies should also explore how to develop edible caterpillar rearing programs by integrating socio-economic aspects. The commercial potential of edible caterpillar farming has already been demonstrated in terms of the exceptional opportunities it offers. However, the profitability of caterpillar farming is far from being confirmed. We look forward to future research focusing on elucidating and quantifying the costs and benefits that will accrue from edible caterpillar farms. 

Finally, in the face of malnutrition and food shortages (current and predicted), which mainly characterize developing countries, the breeding of edible caterpillars could constitute a new source of sustainable food by addressing various expectations connected to the food security of local populations, thus contributing to the objectives of sustainable development.

## Figures and Tables

**Figure 1 insects-13-00886-f001:**
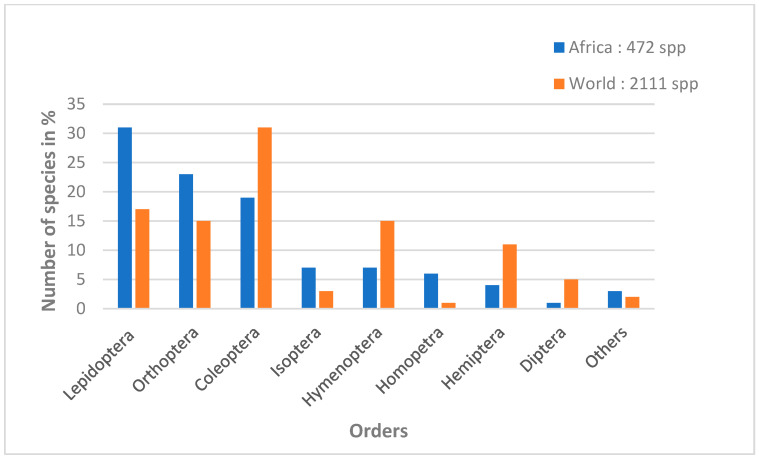
Percentage of insects consumed by order in Sub-Saharan Africa and worldwide (adapted from Mariod [28]).

**Table 1 insects-13-00886-t001:** Classification of the scientific names mentioned in the manuscript.

Branch	Order	Family	Genus/Specie
*Arthropoda*	Coleoptera	Dryophthoridae	*Rhynchophorus ferrugineus* (Olivier 1791)
			*Rhynchophorus palmarum* (Fabricius 1801)
		Tenebrionidae	*Tenebrio molitor* (L. 1758)
	Hymenoptera	Apidae	*Apis melifera* (L. 1758)
	Isoptera	Termitidae	*Macrotermes natalensis* (Haviland 1898) *Macrotermes subhyalinus* (Rambur 1842)
	Lepidoptera	Noctuidae	*Aegocera rectilinea* (Boisduval 1836)
		Bombycidae	*Bombyx mori* (L. 1758)
		Sphingidae	*Clanis bilineata* (Walker, 1866)
		Danainae	*Danaus plexippus* (L. 1758)
		Notodontidae	*Anaphe infracta* (Boisduval 1847)
			*Anaphe venata* (Butler 1878)
			*Antheua insignata* (Gaede 1928)
			*Elaphrodes lactea* (Gaede 1932)
		Lasiocampidae	*Gonometa postica* (Walker 1855)
			*Gonometa rufobrunnea* (Aurivillius 1992)
		Noctuidae	*Aegocera rectilinea* (Boiduval 1836)
			*Heliothis zea* (Boddie 1850)
		Saturniidae	*Bunaea alcinoe* (Stoll 1780)
			*Cirina forda* (Westwood 1849)
			*Cinabra hyperbius* (Westwood 1881)
			*Cirina Butyrospermi* (Vuillet 1911)
			*Gonimbrasia cocaulti* (Darge and Terral 1993)
			*Gonimbrasia krucki* (Hering 1930)
			*Gonimbrasia zambesina* (Walker 1865)
			*Gynanisa nigra* (Klug 1836)
			*Imbrasia ertli* (Rebel 1904)
			*Imbrasia belina* (Westwood 1849)
			*Imbrasia dione* (Fabricius 1793)
			*Imbrasia epimethea* (Drury 1773)
			*Imbrasia obscura* (Butler 1878)
			*Imbrasia oyemensis* (Rougeot 1955)
			*Imbrasia truncata* (Aurivillius 1909)
			*Lobobunaea phaedusa* (Drury 1782)
			*Samia ricini* (Drury 1773)
			*Tagoropsis flavinata* (Walker 1865) *Usta terpsichore* (Maassen and Weymer, 1885)
	Odonate	Libellulidae	*Orthetrum sabina* (Drury 1770)
	Orthoptera	Acrididae	*Locusta migratoria* (L. 1758)
		Gryllidae	*Acheta domesticus* (L. 1758)
			*Brachytrupes membranaceus* (Drury 1770)
			*Gryllus bimaculatus* (De Geer 1773)
		Tettigoniidae	*Ruspolia differens* (Serville 1838)
*Cyanobacteria*	Oscillatoriales	Phormidiaceae	*Arthrospira platensis* (Gomont 1892)
*Plantae*	Fabales	Fabaceae	*Acacia auriculiformis* (A. Cunn. Ex Benth. 1842)
			*Acacia. Mearnsii* (De Wild 1925)
			*Albizia antunesiana* (Harms)
			*Amphimas pterocarpoides* (Harms)
			*Burkea africana* (Hook)
			*Colophospermum spp*
			*Erythrophleum africanum* (Welw. ex Benth.) Harms
			*Erythrophleum suaveolens* (Guill. and Perr.) Brenan
			*Isoberlinia* (Craib and Stapf ex Holland)
			*Piptadeniastrum africanum* (Hook.f.) Brenan
	Ebenales	Sapotaceae	*Autranella congoensis* (De Wild.) A. Chev. en RCA
			*Vitellaria paradoxa* (C.F. Gaertn).
	Caryophyllales	Nyctaginaceae	*Boerhavia diffusa* (L.)
	Malpighiales	Phyllanthaceae	*Bridelia micrantha* (Hochst.) Baill
	Gentianales	Rubiaceae	*Crossopteryx febrifuga* (Afzel. ex G. Don) Benth
		Apocynaceae	*Funtumia africana* (Benth.) Stapf
	Sapindales	Burseraceae	*Dacryodes edulis* (G. Don) H.J. Lam
		Meliaceae	*Entandrophragma cylindricum* (Sprague and Hoyle)
	Lecythidales	Lecythidaceae	*Petertianthus macrocarpus* (P. Beauv.) Liben
	Agaricales	Psathyrellaceae	*Psathyrella tuberculata* (Pat.) A.H. Sm
	Malpighiales	Phyllanthaceae	*Uapaca guinensis* (Müll.Arg)

**Table 2 insects-13-00886-t002:** Life cycle of Lepidoptera for which the larvae are consumed in Africa.

Species	Duration (Days)				Total Cycle Time	Type of Development
	Egg	Larva	Nymph	Adult		
*Anaphe infracta*	/	56	77	7	140	Monovoltine
*Aegocera rectilinea*	≈3	≈20	≈12	8	≈36	Multivoltine
*Bunea alcinoe*	/	/	/	/	/	Multivoltine
*Cirina forda*	≈35	≈50	270	≈2	357	Monovoltine
*Cirina butyrospermi*	30	33	330	5	398	Monovoltine
*Elaphrodes lactea*	60	60	30	210	360	Monovoltine
*Imbrasia belina*	10	42	210	3	265	Bivoltine
*Imbrasia obscura*	/	/	/	/	/	Monovoltine

Source: [36,37,38,39,40,41,42].

**Table 4 insects-13-00886-t004:** Amino acid composition of caterpillars consumed in sub-Saharan Africa (in mg/g dry matter basis).

Amino Acids	*Cirina butyrospermi*	*Cirina forda*	*Imbrasia epimethea*	*Imbrasia belina*	*Imbrasia oyemensis*	*Imbrasia obscura*	*Imbrasi truncata*
Histidine	23–41.5	32–54	20	17–31.1	2	20	17.4
Lysine	25.3–61.3	56–71	50–74.2	36–74.2	3.3	33	79
Leucine	40.8–59.3	63.1–74	48.2–81	35–82.2	3.3	33	73.1
Isoleucine	31.1–52	18–46.4	29–36	22–75.4	2.4	24	24.2
Methionin	16	18.2–23	12–22.4	9–22	1.1	11	22.2
Phenylalanin	34	43.4–55	45.1–65	25–52	3.2	32	62.2
Proline	32–78	46–75.4	/	44–51	4	35	21.4
Serin	17–126.1	42–53	/	44–56.2	3	28	49
Threonin	23.1–138.1	56.4–61	34.1–48	27–73	3	29	47
Tyrosine	16	43.1–75	75	37–64	4.1	41	77
Tryptophan	/	23–73.1	16	7–12	1	10	17
Valine	9–63.4	49.4–66	44–102	32–57	3	27	102

Source: [28,41,43,52,58,74,75,76,77,78,79,80,81,82].

**Table 5 insects-13-00886-t005:** Fatty acid composition of caterpillars consumed in sub-Saharan Africa (% on dry matter basis).

Fatty Acids	*Cirina butyrospermi*	*Cirina forda*	*Imbrasia epimethea*	*Imbrasia belina*	*Imbrasia oyemensis*	*Imbrasia obscura*	*Imbrasia truncata*
Lauric acid (C12:0) SFA	0.08	<0.1	0.2	<0.1	1.66	/	trace
Myristic acid (C14:0) SFA	0.3–0.6	0.7	0.6	0.3–1.2	0.5–1.9	0.2	0.2–0.3
Pentadecanoic acid (C15:0) SFA	/	<0.1–1.9	trace	0.1	/	0.2	trace
Palmitic acid (C16:0) SFA	17.9–27.5	7.4–13	23.2	3.2–31.9	10.1–46	17.1	22.3–20.6
Palmitoleic acid (C16:1) MUFA	0.3	0.2–4.3	0.6	0.1–1.8	/	0.3	0.2–0.5
Margaric acid (C17:0) SFA	0.1–1.3	1.1–5.8	/	0.3–0.4	/	1.1	1.17
Heptadecenoic acid (C17:1) SFA	/	/	/	0.12	/	/	/
Steric acid (C18:0) SFA	35.4–8.9	6.9–16	22.1	1.7–13.5	7.2	18–38.5	16.4–22.3
Oleic acids (C18:1) MUFA	0.4–26.4	3.7–17.9	8.4	1.6–34.2	34.6	8–40.3	7.4–9.5
Linoleic acid (C18:2 n6) MUFA	4.5–30.2	8.1–29.2	7	1.6–10.9	11.2	6.6–9.2	7.1–7.6
α-linolenic acid (C18:3 n3) MUFA	0.8–35.8	4.9–45.3	35.1	19.6–29.4	/	0.8–41.1	28.6–42.6
Arachidic acid (C20:0) MUFA	0.4	2.4	<0.1	0.3–0.4	/	0.3	0.3–8.7
Eicosadienoic acid (C20:2) PUFA	/	0.2	/	/	/	/	0.5
Arachidonic acid (C20:4) PUFA	/	<0.1	<0.1	0.5	/	0.3	0.3
Eicosapentaenoic acid (C20:5) PUFA	/	<0.1	/	/	/	/	
Docosenoic acid (C22:1) MUFA	/	/	/	/	/	/	0.1

Source: [28,41,43,44,49,50,58,62,68,75,76,77,78,79,81,82,89,90].

**Table 6 insects-13-00886-t006:** Mineral composition (mg/100 g dry matter basis) of caterpillars consumed in Sub-Saharan.

Minerals	Ca	K	Mg	P	Na	Fe	Zn	Mn	Cu
*Cirina butyrospermi*	32–0.2	1160–1278	150–169	390	13	13–31	1.9–10	0.6–10	0.13
*Cirina forda*	7–634	48–2130	296–1180	46–1090	42–128	1.3–64	3.7–24.2	6.4–7.5	/
*Imbrasia epimethea*	225	1258	402	666	75	13	11.1	5.8	1.2
*Imbrasia belina*	174	1 032	160	543	1024	31	14	3.9	0.91
*Imbrasia oyemensis*	73	680	610	514.01	730	70.214	11.185	387.9	387.9
*Imbrasia obscura*	100	970	12–240	280	20	/	/	/	/
*Imbrasia truncata*	122–132	1250–1348	178–192	841	170–183	8.1–8.7	10.3–11.1	3.2	1.3–1.4
Recommended Daily Intake	900	420	420	750	1500	9	11	/	1.5

Source: [41,52,62,75,76,79,82].

**Table 7 insects-13-00886-t007:** Availability of edible caterpillars and their host plants.

Scientific Name	Availability	Food Behavior	Host Plants	References
*Anaphe infracta*	January to December, August to September	Polyphagous	*Bridelia minrantha, Cynometra alexandri Triumfetta manrophylla Alhizzia fnstiyiata*	[100]
*Bunaea alcinoe*	June to August	Oligophage	*Pycnanthus angolensis*, *Dacryodes edulis*	[99,101]
*Cirina forda*	June to August, September to December	Polyphagous	*Albizia antunesiana, Erythrophleum suaveolens*, *Baillonella toxisperma*, *Burkea africana*, *Crossopteryx febrifuga*, *Erythrophleum africanum*, *Vitellaria paradoxa*	[99,102,103]
*Cirina butyrospermi*	June to August	Monophage	*Vitellaria paradoxa*,	[37,101,104]
*Elaphrodes lactea*	December to January	Oligophage	*Albizia ferruginea*	[99]
*Imbrasia belina*	December to January, April to May	Monophage	*Colophospermum mopane*	[35]
*Imbrasia obscura*	October to February	Monophage	*Pentaclethra macrophylla*	[99]
*Imbrasia epimethea*	June to September, January to February	Oligophage	*Funtumia africana*, *Dacryodes edulis*	[99,101,103]
*I. truncata*	August	Oligophage	*Amphimas pterocarpoides*, *Petertianthus macrocarpus*, *Piptadeniastrum africanum*	[75,103]
*I. oyemensis*	June to August, August to September	Monophage	*Entandrophragma cylindricum*	[101]

## Data Availability

Not applicable.

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
