# Peer review of "Human Consumption of Insects in Sub-Saharan Africa: Lepidoptera and Potential Species for Breeding"

_insects, 2022, doi:10.3390/insects13100886_

Round 1

Reviewer 1 Report (Previous Reviewer 1)

line119 and elsewhere: Include the full name of genus, at first instance

Author Response

Reviewer 2 Report (New Reviewer)

The manuscript “Human consumption of insects in Sub-Saharan Africa: Lepidoptera and potential species for breeding” submitted for review covers an overview of caterpillar life cycle, nutritional composition and availability associated to specific host plants. The manuscript is interesting, timely and well written.

 However, some minor aspects should be addressed to be suitable for publication.

1. In keywords, replace “host plant1” for “host plant”

2. In figure 1 mention in the respective caption if is “adapted from”.

3. Tables throughout all the document are not found after their mention in the text. Please add the appropriate tables in the correct place.

4. Line 115: table number is missing

5. In table 2, are the data presented on a dry matter basis? If so, please add this information

6. In section 4, please add the meaning of the acronym DRC.

7. In section 5, please replace “revealed the presence of oxalate (15.47±1.88 mg/100 g), (15.47±1.88 mg/100 g),” for “revealed the presence of oxalate (15.47±1.88 mg/100 g)”. The data is repeated.

8. In References section, the reference numbers appear in duplicate. Please check this section.

Author Response

This manuscript is a resubmission of an earlier submission. The following is a list of the peer review reports and author responses from that submission.

Round 1

Reviewer 1 Report

Introduction: By 2050, the world's population will exceed 9 billion, with a significant impact on global food production.

Second paragraph (lines 5-6): change from French to English

Table 7: heading change to English

Good luck

Reviewer 2 Report

The manuscript is interesting and the suggested focus (Lepidoptera in Africa for human consumption) is very nice. 

It is however long and should be shortened by excluding less relevant information (for instance on non-lepidoptera) and by writing more concisely.

Moreover, more information on rearing, and cycli of the most important species should be included. This should be coupled with clear suggestions on their propagation in practice and determining gapsin knowledge. From this concrete suggestions for further studies should be derived. I have included many small suggestions in the PDF. 

Some French sentences were present; was this already published and is this a translation? Moreover, check where italics should be used and where not.

Round 2

Reviewer 2 Report

It seems most of my comments were addressed, but not all. Please go through them again.

The revised version, without a rebuttal, makes it difficult to quickly see the changes made. Please write a rebuttal letter.

Moreover share a manuscript version in which all changes made can be seen instead with comments saying "Changed".

The sections that have been added are written very loosely "worms". Please pay attention to those addittion as I would consider those the difference between accepting or rejecting.

Round 3

Reviewer 2 Report

Please find some comments in the PDF. Yellow markings indicate errors, for instance the lack of italics where needed.
